# Molecular Mechanisms to Target Cellular Senescence in Hepatocellular Carcinoma

**DOI:** 10.3390/cells9122540

**Published:** 2020-11-25

**Authors:** Constanze Mittermeier, Andreas Konopa, Susanne Muehlich

**Affiliations:** 1Cancer Science Institute of Singapore, National University of Singapore, Singapore 117599, Singapore; constanze.mittermeier@nus.edu.sg; 2Department of Chemistry and Pharmacy, Molecular and Clinical Pharmacy, Friedrich-Alexander-Universität Erlangen-Nürnberg, 91058 Erlangen, Germany; andreas.konopa@fau.de

**Keywords:** senescence, HCC, SRF, DLC1, MRTF, senolytics

## Abstract

Hepatocellular carcinoma (HCC) has emerged as a major cause of cancer-related death and is the most common type of liver cancer. Due to the current paucity of drugs for HCC therapy there is a pressing need to develop new therapeutic concepts. In recent years, the role of Serum Response Factor (SRF) and its coactivators, Myocardin-Related Transcription Factors A and B (MRTF-A and -B), in HCC formation and progression has received considerable attention. Targeting MRTFs results in HCC growth arrest provoked by oncogene-induced senescence. The induction of senescence acts as a tumor-suppressive mechanism and therefore gains consideration for pharmacological interventions in cancer therapy. In this article, we describe the key features and the functional role of senescence in light of the development of novel drug targets for HCC therapy with a focus on MRTFs.

## 1. Introduction

Hepatocellular carcinoma (HCC) is a malignant cancer of liver cells. It is the sixth most common cancer, accounting for around 5% of all cancers in the world [1]. With 745,000 registered deaths per year, HCC represents the second common cause of cancer deaths worldwide [2,3]. Furthermore, the incidence of HCC has significantly increased since the 1980s, and hit in 2018 an estimated annual incidence of around 841,000 new cases globally [1,2,4]. The disease predominantly affects the male population compared to the female population with a ratio of 4:1 [5]. HCC has an average five-year survival of <15% and the high rate of tumor recurrence contributes to the poor outcome in the majority of patients with HCC [6,7,8]. The etiopathogenesis of HCC is linked to hepatitis B (HBV) and hepatitis C (HCV) viral infection, alcohol consumption and aflatoxin B1 contamination, triggering chronic liver injury and cirrhosis. During this transition, hepatocytes lacking telomerase activity exhibit progressive telomere shortening and DNA damage [9]. As a consequence, oncogenes and cyclin dependent kinase (CDK) inhibitors such as p16^INK4a^ and p21^Cip1^ are activated to induce a senescence barrier in the preneoplastic cirrhosis stage. Cellular senescence in the liver induces growth arrest in cells at risk of malignant transformation and has mostly been associated with inhibition of HCC growth and progression. In hepatic stellate cells that mainly contribute to extracellular matrix (ECM) production and liver fibrosis, induction of cellular senescence is able to limit liver fibrosis which is considered as the precursor of HCC development [10]. Senescent cells remain metabolically active and secrete cytokines termed “senescence-associated secretory phenotype” (SASP) factors. As a result, cellular senescence inhibits HCC development by prompting clearance of hepatic stellate cells and hepatocytes through a concerted action of innate and adaptive immunity [10,11]. SASP factors can reinforce the senescence response in an autocrine and paracrine fashion [11]; however, SASP factors can also have pro-proliferative effects on neighbouring cells [12]. Inactivation of major senescence-inducing genes (e.g. p53, p16^INK4a^, p15^INK4b^) enables neoplastic cells to bypass the senescence barrier [13]. Transformed hepatocytes activate hepatic progenitor cell expansion through releases of galectin-3 and alpha-ketoglutarate [14]. The underlying molecular mechanisms for HCC formation, however, are far from being fully understood and therefore, pathways as well as critical factors involved in HCC development and progression remain to be investigated. Given the implications of cellular senescence and SASP effector functions in HCC development, pro-senescence therapies and concomitant alteration of the SASP might provide new pharmacological avenues for an efficient and personalized HCC treatment.

In this review, we summarize the key features and molecular players of cellular senescence in the light of potential interventions and therapy options for HCC.

## 2. Cellular Senescence and Its Molecular Players

Cellular senescence was initially described by Hayflick and Moorhead in 1961 as the fact that normal somatic cells in culture have a limited ability to proliferate and thus, have a finite number of replicative cycles [15,16]. This reduced cell proliferation is explained by telomere shortening. As the DNA polymerase is working only in the 5’ to 3’ direction, it is not able to replicate the lagging strand of the DNA and thus, the cell’s telomeres become shorter with each cell division [17,18]. In contrast to the replicative senescence that relies on shortened telomeres with each cell division, senescence can also occur in cells upon exposure to environmental stress or different DNA damaging agents, such as reactive oxygen species (ROS) or chemical and biological mutagens, inducing oxidative stress or DNA damage. According to its origin, it is termed stress-induced premature senescence (SIPS) [19,20,21]. Although the senescence markers and players of cells undergoing SIPS as well as replicative senescence differ, both senescent cell types activate the DNA damage repair (DDR) in response to the induced senescence (see Table 1). The phenomenon of cellular senescence is observed both in pathological as well as in physiological processes such as injury, aging or cancer [22,23]. Moreover, it plays a crucial role in fibrosis, embryogenesis, tissue repair and tissue remodeling [24,25,26].

In 1997, Serrano and colleagues described a novel and different form of senescence: the oncogene-induced senescence (OIS) [27]. This OIS is characterized by a cell cycle arrest elicited by the overexpression of activated oncogenes or by the loss of tumor suppressor genes (see Table 1) [28,29,30,31].

Further studies demonstrated that a mitogenic stimulation of cells is required for a senescence response as rodent cells were not able to become senescent in serum-free medium [32,33]. The non-proliferative state of senescent cells acts as a naturally occurring response of cells against stress and can inhibit the proliferation of potentially malignant cells due to a cell cycle arrest. Therefore, the concept of senescence activation emerged as a potential tumor suppressor mechanism [27]. Senescence induction and thus, its tumor suppressing function is driven by many various pathways and a multitude of different stimuli. More than 50 oncogenes that are able to induce OIS are known, thereby underlining the complexity of senescence induction [34]. For example, oncogenic H-Ras and other members of the Ras signaling pathway, such as Braf, Mek and Raf were described to cause an OIS response [30,35,36]. Additionally, the overexpression of different oncogenes, such as EGFR, HER2 and PI3K, was shown to drive tumor cells into senescence [37,38]. The p16^INK4a^/pRb pathway downstream of the small GTPase Ras was revealed as the most relevant pathway and p53 as the second important tumor suppressor pathway for senescence induction and proliferation arrest via the activation of the mitogen-activated protein kinase (MAPK) cascade [35,36,39,40]. On the molecular level, the cyclin dependent kinase (CDK) inhibitor and tumor suppressor protein p16^INK4a^ inhibits the CDK4/6 activity and thereby elicits a G1 cell cycle arrest that consequently prevents the phosphorylation of the retinoblastoma (Rb) protein [27,41,42]. As a result, Rb is maintained in its activated, hypophosphorylated state that negatively influences the cell cycle progression from G1 to S phase and thereby causes senescence induction [43,44,45]. Besides hypophosphorylation of Rb, histone H3 methylation in senescence-associated heterochromatin foci (SAHF) is another characteristic feature of senescent cells [46]. This H3K9me3 mark is essential for senescence and requires constant renewal at distinct target gene promoters due to histone turnover [47]. For information on the important topic of the dynamic nature of senescence in cancer, the reader is referred to an excellent recent review [48].

## 3. Role of Serum Response Factor (SRF) and Its Coactivators Myocardin-Related Transcription Factors A and B (MRTF-A and -B) in HCC

In recent years, the role of Serum Response Factor (SRF) and its coactivators Myocardin-Related Transcription Factors A and B (MRTF-A and -B) in HCC and cellular senescence has attached considerable importance. SRF, a transcription factor that governs fundamental biological processes such as cell migration, cell growth, cytoskeletal organization and differentiation in concert with its coactivators MRTF-A and -B has been shown to trigger HCC formation [49,50,51,52,53]. Furthermore, therapeutic knockdown of MRTFs abolishes tumor growth in HCC cells lacking the tumor suppressor Deleted in Liver Cancer 1 (DLC1) in vitro and in vivo by inducing OIS [54,55]. DLC1 is a Rho GTPase-activating protein (RhoGAP) that is heterozygously deleted in 50% of liver cancers [56]. Given the frequency of DLC1 loss in liver cancer, unravelling the signaling cascades initiated by DLC1 appears to be an important task to tackle. Work from our laboratory revealed that DLC1 loss leads to nuclear localization and activation of MRTFs [56]. As shown in Figure 1, the loss of DLC1 intensifies RhoA activity [57]. RhoA activation upon DLC1 loss results in actin polymerization, releases MRTFs from binding to monomeric G-actin and enables MRTFs to translocate to the nucleus and to activate gene transcription through SRF binding [58,59]. We found that the actin-binding protein Filamin A (FLNA) is a novel interaction partner of MRTF-A and links changes in actin polymerization to transcriptional activity of SRF [60,61,62]. The nuclear export of MRTF-A is facilitated by phosphorylation of MRTF-A and its direct binding to G-actin [63,64,65]. Impairment of MRTF nuclear localization by the newly identified small molecule NS8593 reduces MRTF/SRF-dependent target gene expression and HCC cell proliferation in vitro and in vivo by inducing OIS [54,60,63]. NS8593 is a negative gating modulator of the transient receptor potential cation channel TRPM7 that plays a pivotal role in cell proliferation, survival and development [66,67]. Reminiscent of the effects of MRTF depletion, pharmacological blockade of TRPM7 by NS8593 provokes growth arrest of HCC xenografts by OIS, characterized by elevated p16^INK4a^ expression and Rb hypophosphorylation [54,68]. Another inhibitor of the SRF/MRTF pathway that impairs nuclear localization of MRTF-A is the small molecule CCG-1423 [69]. At present, it is not known whether CCG-1423 affects HCC growth and OIS; however, CCG-1423 has recently been shown to revert the hyperproliferation of Hodgkin Lymphoma (HL) in vitro and in vivo [70]. CCG-1423 inhibits the atypical actin-regulatory protein MICAL-2 and reduces nuclear localization of MRTF-A and MRTF/SRF-dependent gene expression by increasing nuclear G-actin levels [69]. Consistent with this, it has been shown that nuclear activity of MRTF-A is regulated by actin dynamics in the nucleus [63,71].

Several lines of evidence suggest that alterations in the actin cytoskeleton controlled by MRTFs actively participate in the decision of cell proliferation versus senescence. Besides nuclear accumulation of monomeric G-actin, alterations in the actin cytoskeleton linked to cellular senescence include dephosphorylation of the actin-disassembling factor cofilin-1 [72]. Cofilin-1 was also recently described to tightly control the turnover and dynamics of the nuclear F-actin filaments in the daughter cell nuclei after mitotic cell division [73]. Consistent with the fact that phosphorylation of cofilin by LIM kinase (LIMK) 1 prevents the depolymerization of F-actin, inhibition of LIMK 1 activity was observed in senescent cells [72,74]. Furthermore, Rho-associated kinase 1 (ROCK1) expression was significantly reduced in senescent cells [75]. ROCK1 and 2 have extensively been studied as major downstream effectors of RhoA, playing a central role in the coordination of actin dynamics [76,77]. Another activator of MRTF/SRF signaling with an arising role in cancer progression is microtubule-associated serine/threonine kinase-like (MASTL) [78,79,80,81]. Mechanistically, MASTL associated with MRTF-A and supported its nuclear retention and transcriptional activity [78]. On a functional level, MASTL promoted contractile actin stress fibers and actomyosin contraction by expression of several MRTF/SRF target genes [78]. Therefore, a RhoA/ROCK/actin-dependent mechanism emerges for MRTF/SRF-directed control of transcription in the nucleus and regulation of cellular senescence.

The effects of MRTFs on cellular senescence are mediated by the transmembrane protein Myoferlin (MYOF) via the Ras/MEK/ERK and p16/Rb pathways [82]. Ras activation is facilitated by the activation and phosphorylation of the epidermal growth factor receptor (EGFR) upon MYOF knockdown in HCC cells and xenografts [82,83]. MYOF is involved in the stabilization of different receptor tyrosine kinases, such as the insulin-like growth factor (IGF) or the vascular endothelial growth factor (VEGF) receptors, as well as in the modulation of receptor recycling or the degradation rate [84,85,86]. The finding that MYOF depletion impairs the EGFR degradation and thereby provokes OIS, reflects that the endocytotic activity, and especially the receptor-mediated endocytosis, is significantly decreased in senescent cells [87,88] and explains why senescent cells do not respond properly to external stimuli by growth factors, such as EGF [89]. In cancer cells lacking MYOF, their invasive capacity is reduced due to a reversion of the epithelial-mesenchymal transition (EMT), the mesenchymal-epithelial transition (MET) [82,90]. The impaired invasive potential of MYOF depleted senescent cells also results from a significant decrease of matrix metalloproteinase (MMP) production and secretion that promote tumor spread within fibrotic liver tissues [86,91,92]. Correspondingly, HCC is associated with an increased MMP2 expression [93]. Additionally, the NOD-like receptor X1 (NLRX1) inhibits the EMT and the cell’s invasiveness by repressing the PI3K-AKT pathway in HCC cells [94]. Because the EMT represents a critical process for HCC progression and NLRX1 overexpression is also associated with the induction of senescence it serves as a tumor suppressor in HCC [94]. Taken together, targeting MRTFs and SRF and its target genes such as MYOF represent promising therapeutic options for HCC by inducing cellular senescence and senescence-associated alterations in the cytoskeleton that cease HCC cell migration and proliferation.

## 4. Senescence and Fibrosis

As senescent cells are known to increase their number with age and have been implicated in several age-related degenerative disorders mainly via the secreted proteins from their SASP, cellular senescence plays an important role in tissue remodeling and fibrotic diseases [95,96]. Fibrosis can occur in many different tissues, such as liver, lung or heart, and is a consequence of dysfunctional tissue maintenance and regeneration due to connective tissue deposition and an accumulation of extracellular matrix proteins leading to permanent scar tissue [97,98,99]. There is emerging evidence that cellular senescence contributes to the development of idiopathic pulmonary fibrosis (IPF), a progressive lung disorder showing a damage of lung structure and function [100,101,102]. The molecular mechanisms driving pulmonary fibrosis remain largely uncharacterized and treatment options are currently limited. MRTFs play a key role in lung fibrosis. During the fibrotic process, MRTF-A’s and -B’s translocation into the nucleus and activation of pro-fibrotic target genes via the RhoA-actin signaling axis takes place [103]. Inhibition of the nuclear accumulation of MRTF-A in lung fibroblasts has been shown to decrease lung fibrosis comparable to a global deletion of MRTF-A in a murine model of fibrosis induced by bleomycin [104]. The involvement of senescence in the pathogenesis of IPF is in line with the finding that IPF is a disease of aging and mainly occurs in older individuals [105,106,107,108]. Several different types of cells, such as fibroblasts or epithelial cells, show a senescent phenotype in IPF lungs [109,110,111] accompanied by enhanced expression of established senescence markers, such as p16 and p21 [100,111,112]. Sustained activation of Wnt/β-catenin signaling as well as an aberrant expression of the stem cell marker Nanog were found in senescent IPF fibroblasts [100]. Furthermore, senescent cells secrete elevated amounts of interleukins (IL), including IL-1β, IL-6 and IL-8, which can contribute to the differentiation of fibroblasts into myofibroblasts [113]. Underlining the fact that the development and progression of fibrosis are driven by the accumulation and presence of senescent cells, many studies revealed an anti-fibrotic effect due to the elimination of senescent cells that in turn could improve the lung structure and function in aged mice having lung fibrosis and reverse pulmonary fibrosis [111,114,115]. Importantly, the clearance of senescent cells by senolytic therapies is able to prevent fibrosis and improves the symptoms of fibrosis patients representing a promising treatment option [116,117]. Similarly, senescent cells accumulate in liver fibrosis and cirrhosis, an advanced stage of liver disorder developing from liver fibrosis [118]. HCC is known to be strongly associated with liver fibrosis and cirrhosis, as about >80% of HCCs arise in patients with hepatic fibrosis or cirrhosis and approximately one in three patients with liver cirrhosis will develop HCC [7,119]. Since chronic liver injury triggers inflammatory and wound-healing processes and in consequence hepatic fibrosis that may progress to cirrhosis if not treated successfully, it essentially contributes to hepatocarcinogenesis [119,120,121]. There is mounting evidence that SRF and MRTFs play an important role in liver fibrosis as HCC tumor tissue derived from livers of SRF-VP16^iHep^ mice expressing a constitutively active variant of SRF shows a stronger fibrotic microenvironment with increased collagen depositions than the precancerous nodular tissue [122]. From this murine HCC model, a crucial function of miRNAs in controlling liver fibrosis was recently uncovered [122]. A network of miRNAs was revealed to regulate remodeling, signaling and structural components of the fibrotic extracellular matrix (ECM) [122]. ECM remodeling triggers a positive feedback loop that leads to nuclear translocation of MRTFs and the transcriptional coactivator Yes-associated protein (YAP), resulting in increased expression of MRTF/YAP dependent target genes [123]. Several MRTF target genes, such as α-smooth muscle actin (α-SMA) and connective tissue growth factor (CTGF), are involved in fibrosis [104,124,125]. Activation of MRTFs [126] and MRTF dependent target gene expression (e.g. α-SMA, CTGF) induce the transition of fibroblasts to myofibroblasts that are responsible for the deposition of extracellular matrix components being observed in fibrosis [127,128,129]. Given that targeting or genetic deletion of MRTF-A disrupted collagen synthesis and deposition and reduced fibrosis in different organs such as lung and kidney [130,131], MRTF inhibitors may open up new opportunities for antifibrotic therapies and avoid drug resistance development caused by ECM deposition. Consistently, the novel MRTF inhibitor CCG-222740 prevented scar tissue formation in a preclinical fibrosis model [132]. The related small molecule CCG-203971 targeting myocardin and MRTF-A resulted in the inhibition of hepatic stellate cells (HSCs) activation, exhibiting a major event in liver fibrosis, by directly inhibiting SRF and MRTF nuclear translocation [133]. Because the authors of the study observed no effect of CCG-203971 on markers of inflammation, a direct effect of the inhibitor on HSCs and liver fibrosis is proposed [133]. These data suggest that targeting the MRTF/SRF signaling pathway represents a promising therapeutic approach to treat liver fibrosis and to prevent the progression to HCC.

## 5. Senescence Associated Secretory Phenotype (SASP) and HCC Formation

Senescence acts as a potent anticancer mechanism and protects neoplastic cells from malignancies on the one hand, while on the other hand it is considered as a driving force of aging and age-related diseases [134,135,136]. Due to this fact, it is indispensable for the organism to eliminate the senescent cells. The clearance and degradation of senescent cells are accomplished by the immune system [137,138]. The first in vivo evidence for this was obtained from a mosaic mouse model of liver carcinoma, in which RNA interference was used to conditionally regulate the expression of p53 in liver cancer cells [137,139,140]. Restoration of endogenous p53 upon establishment of solid tumors led to senescence induction and tumor regression triggered by a senescence-associated secretory phenotype (SASP) mediated immune clearance of senescent liver tumor cells [137,139,140]. The SASP that represents a characteristic attribute of senescent cells is of particular importance. The major outcome of the SASP is an activation of an immune reaction rather than a reinforcement of the senescence response. The acquisition of a SASP has the ability to turn senescent cells into pro-inflammatory cells that secrete high levels of adhesion molecules, chemokines, cytokines as well as growth factors that can recruit and activate specific immune cells from the innate and also the adaptive immune system [141,142,143]. Hepatocytes can generate an SASP mediated inflammatory environment that recruits CD4^+^ T cells and macrophages in response to NRas activation, resulting in subsequent elimination of senescent cells and impairment of NRas induced hepatocarcinogenesis. [144]. The recognition and elimination of senescent cells are carried out by different cell types, mainly by T cells, monocytes/macrophages and natural killer (NK) cells [145,146,147,148]. Despite the clearance of senescent cells by the immune system, senescent cells accumulate during aging within different tissues [149]. This incomplete cell elimination and accumulation with age may be explained with a decline in immune function and a permanent weakening of the immune system caused by essential changes in the formation and functionality of immune cells during aging [139,140]. As a consequence, the persistence of senescent cells in different tissues and the secretion of SASP factors result in a chronic pro-inflammatory microenvironment and promotes the development of tissue aging and the development of age-related diseases [150,151,152,153] (Figure 2). Here, fibrosis, type 2 diabetes, obesity, cardiovascular diseases, sarcopenia, osteoarthritis and neurological disorders are attributed to age-related diseases associated with increased senescence [22,95]. Obesity has recently been recognized as an important factor driving the development of HCC [154]. Changes in gut microbiota caused by obesity promote the release of bacterial metabolites such as deoxycholic acid (DCA), which in turn induce senescence and release of SASP factors in hepatic stellate cells (HSCs) [155]. DCA-induced SASP factors secreted from HSCs promote obesity-associated HCC development due to an enhanced SASP level [156]. In contrast, CCL2-CCR2 proteins classified as SASP factors support the elimination of senescent hepatocytes and thereby suppress HCC formation [11,144]. While SASP factors are able to suppress liver cancer initiation, they can promote tumorigenesis in advanced stages of HCC development, altered secretory activities of senescent cells can promote tumorigenesis due to their immunosuppressive effect and changes in the tissue microenvironment [157]. Several SASP factors are able to enhance the senescence of neighbor cells. The secretion of the interleukins IL-6 and IL-8, or insulin-like growth factor-binding protein 7 (IGFBP7) increases oncogene induced growth arrest caused by Ras and Braf [158,159,160]. Given the potential implications of the SASP in HCC development, altering the SASP might provide new therapeutic opportunities for the treatment of liver cancer.

## 6. Functional Role of Senescence in Light of the Development of Novel Drug Targets

For tumor therapy, activating the senescence response emerges as an important and promising strategy to curb tumor growth [27,30,35,161] since it is often disabled in cancer cells [162]. OIS maintains the tumor cell in a premalignant and non-aggressive state thereby restricting the further cell growth, whereas cells not inducing a senescence response progressed to a malignant state [28,163,164]. Michaloglou and colleagues demonstrated that human melanocytes with a sustained expression of oncogenic BRAF mutations underwent OIS and remained benign instead of progressing into malignant melanoma [30,165]. In this context, the loss of the tumor suppressor phosphatase and tensin homolog (PTEN) in BRAF-mutated cells promotes tumor progression and the development of metastatic melanoma in vivo [29]. Additionally, Kang et al. demonstrated that the induction of senescence in preneoplastic hepatocytes efficiently prevented HCC [144]. 

Several studies highlighted the importance of novel drugs that induce senescence, such as the CDK4/6 inhibitors Palbociclib, Ribociclib or Abemaciclib showing promising benefits as anticancer compounds [166,167,168,169]. The combination of Palbociclib administration with current breast cancer therapies significantly increased the median progression-free survival in clinical trials [168]. Moreover, the inhibition of Src homology region 2-containing protein tyrosine phosphatase 2 (SHP2) prevented invasion and mammary tumor growth in mice due to senescence induction [170,171].

Directed therapeutic intervention depends on a deep understanding of the signaling pathways and molecular players through which OIS is manifest, as described in the section above. Targeted therapies aimed at the selective enhancement of senescence in liver cancer cells could be used for HCC therapy. This approach differs from conventional therapeutic regimens that affect both normal and cancer cells. At the moment, the kinase inhibitors Sorafenib and Lenvatinib are the only by the Food and Drug Administration (FDA) approved drugs for HCC therapy [172,173,174,175]; however, the arising of primary or acquired resistance against the drugs is a major issue [175,176,177]. Due to these limitations, there is a pressing need to identify novel drugs and targets for HCC therapy. To minimize the possible effects of the accumulation of senescent cells and the secretion of SASP factors, which drive human age-related pathologies, senolytics may be beneficial. Senolytics are pharmacologically active compounds with preferential cytotoxic activity for senescent cells [178,179,180]. Recently it was shown that the loss of the newly discovered HCC suppressor fructose 1,6-bisphosphatase 1 (FBP1) leads to senescence induction in hepatic stellate cells (HSCs) and a related release of SASP, which triggered HCC formation. The use of senolytics such as Dasatinib/Quercetin or Navitoclax (ABT-263) results in the clearance of senescent HSCs and inhibition of HCC progression [181]. Another example is the bromodomain and extra-terminal domain (BET) family protein degrader (BETd) which was identified as a senolytic by high-throughput screening. BETd therapy removes senescent HSCs in adipose mouse liver, thereby inhibiting the progression of liver cancer [182]. Since HCC formation is often associated with SASP release in senescent HSCs, inhibition of SASP release or the reduction of their disease-causing phenotypes as a therapeutic target might be an appropriate approach. This strategy is called senomorphic or senostatic therapy [183]. In comparison to senolytics, senostatics/senomorphics do not eliminate senescent cells, rapamycin for example inhibits the release of SASP via mTor [184]. A dangerous side effect of senomorphics is reduced wound healing due to SASP suppression [185]. Table 2 shows a selection of some SASP inhibitors and senolytics. At the moment, options for eliminating senescent cells for HCC therapy are in a preclinical stage.

Furthermore, Zhu and colleagues reported an upregulation of anti-apoptotic gene sets and negative regulators of apoptosis in senescent compared to nonsenescent cells [116]. Members of this family, such as B-cell lymphoma-extra-large (Bcl-xL), are involved in the regulation of programmed cell death by caspase inhibition. Interestingly, dysregulation of the Bcl-2 family has not only been found in several degenerative diseases but also in different cancers, such as HCCs [199,200,201]. Introduction of siRNAs against Bcl-xL led to a reduced survival rate in senescent cells whereas proliferating cells were not affected [192]. Consistently, the administration of Navitoclax (ABT-263), a Bcl-2, Bcl-xL and Bcl-W inhibitor, led to apoptosis and eradication of cells in a senescent state [116] (Figure 2). This effect could also be observed in vivo in mice [202]. Furthermore, the clearance of therapy induced senescent cells by senolytics, leads to a decreased incidence of cardiac dysfunction as well as a reduced recurrence of cancer and a lower toxicity of chemotherapy [203]. The related Bcl-2 inhibitor ABT-737 (Table 2) also suppressed growth of hepatoma cells in combination with Sorafenib [202]. Importantly, SRF binds to the Bcl-2 promoter in vivo and activates the transcription of Bcl-2, while the loss of SRF and MRTFs impairs expression of antiapoptotic Bcl-2 family members [204,205]. Targeting MRTF activity as pro-senescence therapy may therefore not only induce senescence but also eradicate senescent cells by Bcl-2 downregulation, thereby mitigating the deleterious effects of the SASP. It will be an interesting task to tackle in the future if a combination therapy with MRTF inhibitors such as NS8593 [66] or CCG-222740 [206] and senolytics might improve outcomes in tumor therapy, particularly in HCC [116].

## 7. Conclusions

The increase in the incidence of HCC and the difficulties to treat HCC owing to the current paucity of drugs highlight the pressing need to develop new pharmacological approaches for systemic therapy [207,208]. The induction of senescence represents a novel strategy for the treatment of liver cancer, especially in a one-two punch therapy using a combination of a senescence-inducing drug and a second drug that selectively eliminates senescent cancer cells [209]. The causes underlying this two-step anticancer therapeutic concept rely on the distinct features of cancer-associated senescent cells. Senescence induces not only cell cycle arrest elicited by the overexpression of activated oncogenes or by the loss of tumor suppressor genes and accompanied by alterations in the organization of the actin cytoskeleton, but also a context-dependent senescence-associated secretory phenotype (SASP) that directs their immune-mediated clearance described in this review. While senescence induction acts as a powerful tumor suppressor mechanism, the potential tumor promoting effects of the SASP may drive cancer relapse. Therefore, a pro-senescence cancer therapy and subsequent clearance of senescent cells by senolytics, as outlined in Figure 2 and Table 2, has built excitement as a therapeutic intervention and gains even more impetus for liver cancer as the potentially harmful effects associated with aberrant accumulation of senescent cells may be attenuated. Consistent with this, a recent study demonstrated that the anti-depressant Sertraline as well as mTOR inhibitors kill HCC cells that have been rendered senescent by inhibition of cell division cycle 7-related protein kinase (CDC7) which phosphorylates critical substrates regulating the transition from G1 to S phase [188]. Directed therapeutic intervention depends on a profound understanding of the signaling pathways through which cell cycle arrest in G1 phase and cellular senescence is manifest. Pharmacological inhibition of MRTFs induces proliferation arrest due to oncogene-induced senescence (OIS) and inhibits tumor growth in HCC cells in vivo and in vitro [66]. A RhoA/ROCK/actin-dependent mechanism emerges for MRTF/SRF-directed control of cellular senescence, implying a unique therapeutic opportunity to target senescent cells by inhibiting MRTF transcriptional activity. Characterizing the secretome of liver cancer cells treated with MRTF inhibitors is an important task to tackle since the induction of the SASP represents a double-edged sword for tumor control and the detrimental properties of senescent cells make their elimination therapeutically relevant. As shown in Figure 3, the interplay of a pro-senescence therapy, e.g., MRTF or MYOF siRNA application, and a senolytic therapy, e.g. with ABT-263, represents a promising strategy for HCC treatment by positively effecting the bad outcome of an accumulation of harmful SASP due to senescence induction and turning it into a positive outcome for HCC.

Given the high heterogeneity of HCC, it is likely that the response towards senescence activators as well as senolytic drugs will vary among patients. Therefore, future interventions regarding senescence therapy in HCC will likely be biomarker-dependent and more personalized. Despite only a limited number of clinical trials on senotherapies so far, it is highly conceivable that this treatment strategy will achieve better outcomes and survival rates for HCC patients. Here, the major challenges are the development of more advanced and clinically relevant models, the need for accurate and non-invasive senescent biomarkers that senolytic drugs can specifically target and especially, an increase in the selectivity and specificity of senolytics to reduce potential toxicities. The latter can be optimized by modifying the therapeutic agent that it can be activated by external stimuli. In this context, the drug could be loaded into nanocarriers targeting senescent cells as already described for CD9-targeted delivery of rapamycin using lactose-wrapped calcium carbonate nanoparticles or galacto-oligosaccharide-based nanoparticles utilizing specific β-galactosidase expression of senescent cells [210,211]. The encapsulation of drugs reduces the toxic side effects making them an efficient and selective tool for future HCC senotherapy and might help to facilitate their way to clinical trials although experimental data of applied nanocarrier elimination are still needed.

Overall, senescence induction as well as specific targeting of MRTFs in combination with the identification and application of new and reliable senolytic drugs that clear senescent cells may provide a promising therapeutic modality for personalized HCC therapies in the future.

## Figures and Tables

**Figure 1 cells-09-02540-f001:**
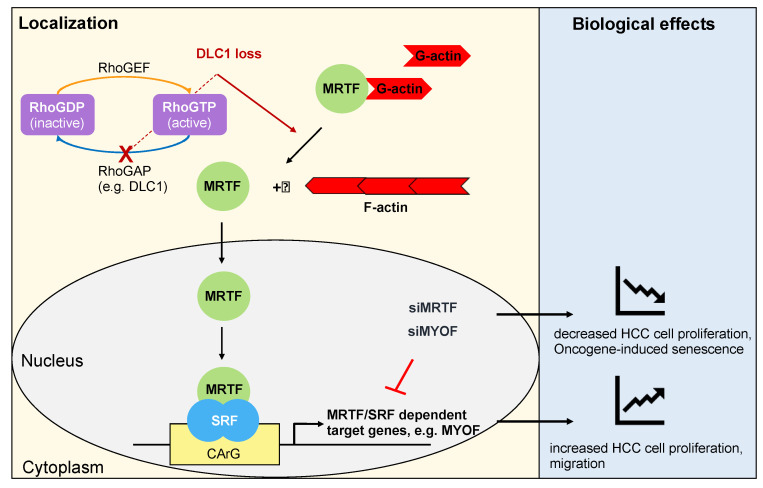
MRTFs and MYOF modulate proliferation and senescence upon DLC1 loss. Loss of the RhoGAP DLC1 results in Rho activation, which leads to actin polymerization, MRTF nuclear localization and expression of MRTF/SRF dependent target genes such as MYOF, resulting in enhanced HCC cell proliferation and migration. Depletion of MRTFs or MYOF inhibits HCC cell proliferation by inducing oncogene-induced senescence.

**Figure 2 cells-09-02540-f002:**
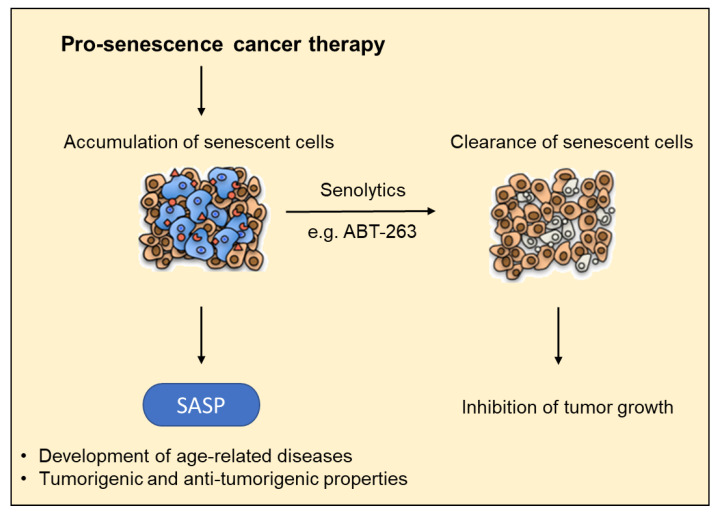
Pro-senescence cancer therapies to combat tumor growth. The accumulation of senescent cells in different tissues due to a reduced clearance as a result of, e.g. a compromised immune system leads to an increase in SASP factor secretion. This favors the development of age-related diseases and may counteract antitumor effects. The combination with senolytics may prevent an accumulation of senescent cells and the resulting deleterious effects of the SASP.

**Figure 3 cells-09-02540-f003:**
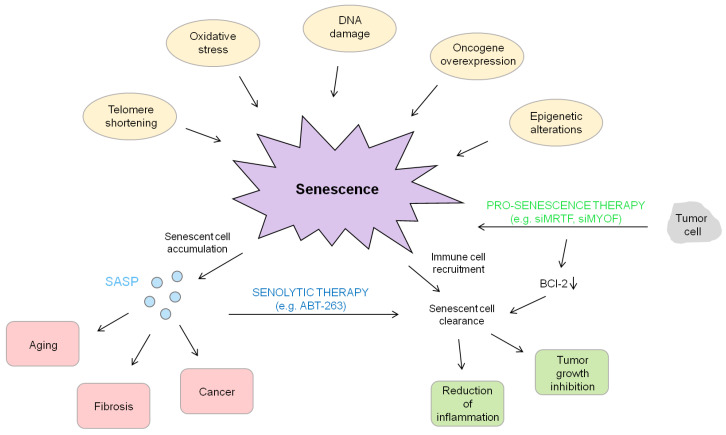
Summary of senescence events. Senescence is activated by a variety of different triggers, such as telomere shortening or oncogene overexpression. On the one hand, an accumulation of senescent cells and the subsequent secretion of SASP can be deleterious by resulting in aging, fibrosis and cancer. On the other hand, the induction of senescence combined with the clearance of senescent cells can be beneficial due to its tumor growth inhibition and suppression.

**Table 1 cells-09-02540-t001:** Senescence markers and molecular players in different types of senescence.

Replicative Senescence	Stress-Induced Premature Senescence	Oncogene-Induced Senescence
Shortened Telomeres: telomeres progressively shorten with each cell division due to end- replication problem 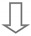 Activation of DDR at telomere ends	Reactive oxygen species (ROS): Oxidative StressDNA Damage 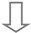 Activation of DDR	Overexpression of oncogenes, e.g., H-Ras, Braf, Raf, HER2 and PI3KLoss of tumor suppressor genes, e.g., PTEN and NF1 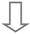 -Activation of p53 pathway via MAPK cascade-Activation of p16^INK4a^/pRb pathway
enlarged, flat morphologymulti-nucleatedincreased β-galactosidase activity at pH 6increased expression of γ-H2Axexpression of cell cycle inhibitors (e.g., p15^INK4b^, p16^INK4a^ and p21^Cip1^)

**Table 2 cells-09-02540-t002:** Drugs targeting senescence in HCC.

Strategy for Senescent Cell Elimination	Drug	Drug Target	References
**SASP Inhibition**	Rapamycin	mTor	[186,187]
	Sertraline	mTor	[188]
	Torin1	mTor	[189]
	Pacritinib	JAK2	[190]
	Ruxolitinib	JAK1/2	[191]
**Senolytics**	Dasatinib	Src kinase, BCR/Abl	[192,193]
	Quercetin	Bcl-xL	[192,193]
	A-1331852	Bcl-xL	[194]
	ABT-263, -737	Bcl-2, Bcl-xL and Bcl-W	[195]
	XL413	CDC7	[188]
	BETd	BRD4/NHEJ	[182]
	Ouabain (cardiac glycoside)	Na+/K+-ATPase	[196]
**Immunotherapy**	Poly(I:C)	NK cell activation	[197]
	CAR-T	uPAR	[198]

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
