# Peer review of "Molecular Mechanisms to Target Cellular Senescence in Hepatocellular Carcinoma"

_cells, 2020, doi:10.3390/cells9122540_

Round 1

Reviewer 1 Report

In this review manuscript, the authors attempted to provide a comprehensive review on the cellular senescence as a molecular target for hepatocellular carcinoma (HCC) therapy.  The topic is of significance and may have an impact on drug discovery and development for HCC.   This reviewer has some suggestions for authors to consider in its revision. \

  1. Cellular senescence is well recognized in the cancer research field. Although a brief historic review may help readers understand the context, the information provided in the first three sections are text-book materials and the three sections should be shortened. In addition, several reviews have been published in last few years, interested readers can be directed to those reviews.
  2. In general, this review does not focus on HCC; the information summarized is general and non-specific.
  3. The section of Senescence and aging does not have a clear linkage to HCC. The same is true for the section of Senescence and the actin cytoskeleton.
  4. A table or figure could be added to illustrate the representative drugs targeting senescence for HCC therapy at the different stage of development.
  5. An in-depth discussion section should be added to illustrate the knowledge gaps between HCC biology and pathology in relation to cellular senescence and drug discovery. Future research directions should also be discussed in greater details.
  6. The majority of references is published before 2015.

Author Response

Dear Reviewer 1,

thank you for considering the revised version of our manuscript cells-971134 titled “Molecular mechanisms to target cellular senescence in Hepatocellular carcinoma” for publication in Cells. We are thankful to the referees and the Editor for pointing out some important modifications needed in the paper. We have thoughtfully taken these comments into account. Please find below our response to the points raised by the reviewers. We appreciated the reviewer’s insightful comments and believe to have addressed all concerns satisfactory.

We have submitted a revised version of our manuscript, with the changes highlighted in red, together with a point-by-point response attached below.

We look forward to hearing from you.

Sincerely yours,

Susanne Muehlich, on behalf of the authors

Reviewer 2 Report

In the reviewed manuscript, the authors described the latest findings on the role of senescence in hepatocellular carcinoma field. Especially, they focused on oncogene-induced  senescence (OIS) mediated by the MRTF-dependent target gene Myoferlin (MYOF) as the underlying molecular mechanism for the growth arrest in vitro and in vivo. Moreover, they have discussed the development of novel drug targets in tumor therapy and their effect on induced senescence in cancer cells.

The manuscript is interesting but should be improved in several points before publication

Please note the general organization of the entire article as well as several individual paragraphs is poor, making it difficult and confusing to follow in places. Second, it is not clear to me what the authors are trying to say in this paper.

Introduction should be rewritten and more focused on mechanisms of hepatocellular carcinoma and very general information about potential link between etiopathogenesis of hepatocellular carcinoma and senescence mechanisms.

Please note that the role of Serum Response Factor (SRF) and its coactivators Myocardin-Related  Transcription Factors A and B (MRTF-A and -B) in HCC should be described as separate section as after II.

In section II, please introduce the information in a new table about senescence markers in hepatocytes. Please compare senescence markers, molecular players in replicative senescence, stress induced premature senescence and oncogene-induced  senescence.

Please changed the title “Senescence and aging “ because in the present form it does not reflect the content of this section. Please note that authors discuss about the Senescence-Associated Secretory Phenotype  and tumorigenesis but not about general aging.

The section entitled “Functional role of senescence in light of the development of novel drug targets” should be presented as one of the last.

Moreover, according to figure 2, drugs induced senescence and  senostatic and senolytics drugs should be better discussed in one common section.

Author Response

Dear Reviewer,

thank you for considering the revised version of our manuscript cells-971134 titled “Molecular mechanisms to target cellular senescence in Hepatocellular carcinoma” for publication in Cells. We are thankful to the referees and the Editor for pointing out some important modifications needed in the paper. We have thoughtfully taken these comments into account. Please find below our response to the points raised by the reviewers. We appreciated the reviewer’s insightful comments and believe to have addressed all concerns satisfactory.

We have submitted a revised version of our manuscript, with the changes highlighted in red, together with a point-by-point response attached below.

We look forward to hearing from you.

Sincerely yours,

Susanne Muehlich, on behalf of the authors

Round 2

Reviewer 1 Report

The authors have addressed all my concerns.

Reviewer 2 Report

The authors have improved manuscript according to my comments